# Parametric Study of Residence Time Distributions and Granulation Kinetics as a Basis for Process Modeling of Twin-Screw Wet Granulation

**DOI:** 10.3390/pharmaceutics13050645

**Published:** 2021-05-01

**Authors:** Timo Plath, Carolin Korte, Rakulan Sivanesapillai, Thomas Weinhart

**Affiliations:** 1Multi-Scale Mechanics, TFE, ET, MESA+, University of Twente, P.O. Box 217, 7500 AE Enschede, The Netherlands; t.weinhart@utwente.nl; 2Process Technology Development, Engineering & Technology, Bayer AG, 51368 Leverkusen, Germany; carolin.korte@bayer.com (C.K.); rakulan.sivanesapillai@bayer.com (R.S.)

**Keywords:** twin-screw wet granulation, residence time distribution, design of experiments, continuous manufacturing, process modeling, virtual prototyping

## Abstract

Twin-screw wet granulation is a crucial unit operation in shifting from pharmaceutical batch to continuous processes, but granulation kinetics as well as residence times are yet poorly understood. Experimental findings are highly dependent on screw configuration as well as formulation, and thus have limited universal validity. In this study, an experimental design with a repetitive screw setup was conducted to measure the effect of specific feed load (SFL), liquid-to-solid ratio (L/S), and inclusion of a distributive feed screw on particle size distribution (PSD) and shape as well as residence time distribution of a hydrophilic lactose/microcrystalline cellulose based formulation. An intermediate sampling point was obtained by changing inlet ports along the screw axis. Camera-based particle size analysis (QICPIC) indicated no significant change of PSD between the first and second kneading section, except for low L/S and low SFL where fines increase. Mean residence time was approximated as a bilinear fit of L/S and SFL. Moreover, large mass flow pulsations were observed by continuous camera measurements of residence time distribution and correlated to hold-up of the twin-screw granulator. These findings indicate fast granulation kinetics and process instabilities for high mean residence times, questioning current standards of two kneading compartments for wet granulation. The present study further underlines the necessity of developing a multiscale simulation approach including particle dynamics in the future.

## 1. Introduction

Industrial wet granulation is presently mainly carried out by batch processing. Recently, the possibility of inline quality testing, real-time release, batch-size flexibility, and reduced footprint, among other advantages, has sparked a growing interest in continuous processing [1,2], especially in the pharmaceutical and bio-pharmaceutical domains [3]. Still, there are barriers for continuous pharmaceutical manufacturing to overcome and crucial challenges in process understanding have to be resolved, such as the control of residence time distributions (RTDs) and process stability [4]. Twin-screw granulation (TSG) is a promising solution for continuous wet granulation. It is flexible in design owing to an adaptive screw geometry with mixing, conveying, as well as kneading elements. Because of a variable throughput, it requires limited scale-up and the granulation rate process is regime-separated [5,6], benefiting both experimental investigations and process understanding. However, the presence of non-conveying screw elements gives rise to back-mixing, which, together with variable powder flowability along the screw axis, underlines the need for an improved understanding of residence time distributions.

Experimental studies have been carried out by many researchers to develop a deeper understanding of the TSG process. Many studies use a design of experiments (DoEs) approach [7,8]. The insights gathered from DoEs allow process optimization and expand the state-of-the-art. However, DoEs provide limited insight for multi-factorial design spaces, and the results depend on the specific screw geometry and formulation [9]. One challenge in setting-up a DoE is to choose the right design space: screw speed, liquid-to-solid ratio (L/S), and throughput are common factors, whereof L/S was repeatedly found to have the most influence on the process. However, throughput and screw speed are correlated. Therefore, in more recent studies, the fill level is used as a factor, reducing the DoE dimensions. The fill level of a TSG can be determined in several ways. Osorio et al. [10] calculated the channel fill level as a fraction of screw capacity, while Gorringe et al. [11] solely takes the fill level of conveying sections into account. However, in both approaches, bulk density plays an important role in calculating the fill level, but bulk density changes throughout the process as the particles agglomerate and grow. A recent study by Meier et al. [12] thus uses the specific feed load (SFL), which measures the mass of particles transported during one revolution of the twin-screw. SFL was found to have a linear correlation with the volumetric fill level and can be calculated by dividing total inlet mass flow (including liquid) $\dot{m}$ by screw speed. In contrast to the dimensionless fill level, SFL has a unit of grams (per rotation). Hence, SFL is only valid for the system it was applied to and is not a suitable parameter for upscaling. However, the advantage of SFL is that it can be calculated directly from experimental data, without estimating bulk density. If the transportation volume of all screw elements is known, the volumetric fill level can also be derived from SFL. Another challenge is aggregating data from different spatial locations along the screw axis. Thus, most experimental studies focused on collecting data at the inlet and outlet of a TSG—see, e.g., [12,13]. Two distinct methods to extract local data have been developed, namely the screw pull-out method [14] and the compartmental method [15]. In the former approach, only a low amount of samples can be gathered from different spatial locations. The latter was designed to overcome sample-size problems by changing the position of the liquid inlet port to collect granule samples at different stages of the granulation process, which further allows to analyse different process stages separately. However, as the particle feeder position is not moved along with the liquid inlet, the method potentially lacks general applicability.

TSG simulations were primarily developed based on these specific DoEs. Alongside, neural-network [16,17] and computational fluid dynamic [18] approaches, as well as population balance modeling (PBM)—see, e.g., [19,20,21]—have been utilized, mainly to model high-shear granulation processes [22]. Most of these developments have been successful in predicting critical quality attributes (CQAs), like the resulting particle size distribution (PSD). Nevertheless, current modeling work as well as experimental studies and measuring devices need improvement in order to determine sufficient correlation between CQAs and critical process parameters, to scale-up methodologies, and to carry out kinetic studies of all steps involved in granulation [1]. In particular, solely solving a PBM is not accounting for the underlying particle dynamics and its interaction with the geometry. This, however, is a crucial factor according to Vercruysse et al. [23] and Portier et al. [9]. Furthermore, appropriate quantities to compare different granulators by means of energy input, residence time, and granulator fill level remain to be developed [5]. Thus, screw configurations must still be empirically adjusted, because the current simulation models are merely valid under certain boundary conditions and for the specific screw configurations for which they have been developed [24]. Therefore, recent approaches have adopted the discrete particle method (DPM, also called DEM), which has proven to efficiently simulate high shear granulation [25,26], in order to include particle dynamics into PBM simulations [27]. DPM can be calibrated from experimental data, which, in contrast, is scarcely possible for PBM because of the kernels’ statistical nature. Passing experimentally calibrated and validated data from DPM to PBM can help overcome those calibration issues and avoid best guesses for PBM kernels. However, a bottleneck of DPM is the small time scales of particle interactions when compared with the PBM time scale, making coupling of DPM to PBM simulations computationally inefficient. Because of the high computational effort, DPM modelling is best applied to smaller volumes, such as simulating different stages of the process independently. Thus, there is a need to develop a comprehensive simulation framework, which includes particle dynamics into PBM and solves the issue of different time scales for simulation. A promising solution is to develop a comprehensive DPM–PBM–CFD heterogeneous multiscale model [28]. The DPM will run in a certain parameter space and save coarse-grained [29] particle properties into a database. Hence, the PBM is able to extract data from this database to calculate mechanistic kernels for simulation. Dynamic programming [30] can be utilized to extend the domain for empirical adjustments as well as to lower experimental efforts. In addition, the insights of this framework can help to develop an application-specific PBM for design optimizations and enable transitioning from real to virtual design spaces.

The focus of this paper is on executing a DoE that is not only designed to optimize a specific screw setup or to get further process insight. Furthermore, it is designed to allow calibration and validation of the proposed simulation framework specifically regarding residence time and particle size predictions. A lactose/microcrystalline cellulose (MCC) based formulation is used, which is known to be very robust in drug development [31]. To utilize the data gathered from our experimental setting for process modeling, it must be as accurate as possible. Therefore, a novel setup based on the idea of Verstraeten et al. [15] has been adapted to gather in-depth knowledge for process modeling with high accuracy. The setup allows to collect data at an intermediate point by moving the inlet ports for liquid and powder to reduce the screw length, and thus will help modeling different sections independently of each other to monitor granulation kinetics. Material characterization, results, and new findings of the conducted DoE will help to calibrate and validate the simulation framework in a later stage of research.

## 2. Materials and Methods

### 2.1. Materials

The investigated lactose/MCC active pharmaceutical ingredient free formulation is listed in Table 1. Tartrazine (Fluorochem, Manchester, UK) was used as water-soluble dye to measure residence time distributions of the liquid via camera, and purified water served as granulation fluid.

### 2.2. Twin-Screw Wet Granulation

Prior to granulation, the premix was blended in a Turbula^®^ T2F mixer (Glen Mills Inc., Clifton, NJ, USA). A Thermo Fisher Pharma 11 extruder system with a TSG conversion kit (Thermo Fisher Scientific, Karlsruhe, Germany) was utilized for the wet granulation experiments. The 11 mm co-rotating continuous granulator has a length to diameter ratio (L/D) of 40:1. The heatable jacket was held constant at 30 °C by a cooler thermostat (Accel 500 LC, Thermo Fisher Scientific, Karlsruhe, Germany). A gravimetric twin-screw feeder (MiniTwin (DDW-) MT, Brabender, Duisburg, Germany) fed the powder into the TSG at a constant solid feed rate of 1 kg/h. Prior to the first kneading section, water was fed into the TSG via a peristaltic pump (Masterflex P/S, Thermo Fisher Scientific, Wesel, Germany) using Marprene tubes with a diameter of 4.8 mm. The screw comprised conveying elements (CEs), kneading elements (KEs), and a distributive feed screw (DFS). The kneading elements are of length 0.25D, with a staggering angle of 60°. The TSG setup without the DFS consisted of the following elements, from engine to outlet (left to right):
0.75D KE − [4D CE (2L/D) − 10D CE (1L/D) − 0.5D CE (0.5L/D) − 1.5D KE − 4D CE (1L/D)]^2^

In experiments where the DFS element was included, it replaces the last CE of the TSG. The above formula lists the length and element type of each section, with the pitch of CEs appended in square brackets. The curly bracket indicates a repetitive part of the screw setup and its power represents the number of repetitions. The first 0.75D KE section served as a spacer. Two pairs of liquid and solid inlets were positioned, one shortly after the engine for long-screw operation (see 3, Figure 1), the other in the middle of the barrel (see 2, Figure 1) for a short-screw operation. Owing to the repetitive screw design, the short screw operation provides the intermediate granule size distribution of the long screw operation. Thus, this design offers insights into granulation kinetics from premix (see 1, Figure 1) to intermediary (short-screw) to the final state (long-screw). Behind the outlet, a conveyor belt (Thermo Scientific Fisher, Pharma 11 Conveyor, Karlsruhe, Germany; Vetter, BK-20-40, Westerstetten, Germany) is placed to allow analysis of the granulated particles for residence time measurement by camera and sample retrieval. See Figure 1 for a schematic overview of the experimental setup. In industrial manufacturing routes, wet granules are dried in a fluidized bed, which produces, for our purpose, undesirable fines. Therefore, to better represent the process, samples were oven-dried (FDL 115, Binder, Tuttlingen, Germany) at 40 °C at ambient humidity for further investigations. A specimen divider (Retsch, PT 100, Haan, Germany) separated samples into smaller divisions for subsequent particle size analysis.

### 2.3. Design of Experiments

A two-level full factorial DoE was carried out to measure the effect of two quantitative parameters, namely SFL and L/S, together with two qualitative parameters, the number of repetitive screw segments (short, long) and the presence of a DFS element at the outlet; see Table 2 for a full overview of parameters. The DoE, therefore, comprised 16 experiments and 4 center points for each combination of the qualitative variables. The center points were executed in triplicate to determine reproducibility. Additionally, four dry experiments (L/S = 0) were performed to assist in calibrating a DPM model. These additional experiments will be excluded from further discussion and published in a later stage of research. A full overview over all experiments, their parameters, and responses can be found in the Table A1.

Pre-studies were performed to identify suitable factor ranges. Originally, it was planned to keep the pitch of all CEs equal. However, below the solid feed section, elements with a higher pitch were needed to allow accurate feeding into the granulator and avoid bridging at the inlet. Application of a DFS element at the outlet of the granulator was also included as factor in the DoE to evaluate its influence on the PSD of the product. Furthermore, it was investigated whether the maximum particle size can be limited by the gap size between two teeth of the DFS.

The DoE was evaluated using the Modde 12.1 software (Sartorius AG, Göttingen, Germany) with the following response parameters: PSD at the in- and outflow and RTD. Multiple linear regression at a significance level of α=0.05 was employed for the evaluation. Backward regression was utilized to obtain the highest model coefficient of prediction (Q^2^), while at the same time maintaining a high coefficient of determination (R^2^). See [32] for further information about Modde.

### 2.4. Particle Shape and Size Distributions

By adopting the methodology described by Verstraeten et al. [15], three different sampling points were obtained:Initial state of particles at the inlet,Intermediary state of particles after one kneading zone (short screw),Final state of particles after two kneading zones (long screw).

The first sampling point represents the data gathered for the initial state of the powder premix. The initial PSD for each particle ingredient was measured only once and considered unchanged for all runs. The main emphasis is on the differences between the second and third sampling points. Therefore, the first sampling point was not included in the results section, but can be found in the data repository. Collection of samples was started after reaching process equilibrium, which was determined by requiring a constant torque.

Distributions of size and shape were measured in triplicate (except for N16 in duplicate) by a QICPIC particle size analyzer (GRADIS, Sympatec, Clausthal-Zellerfeld, Germany) with PAQXOS 4.0 software. The EQPC diameter was used, i.e., the particle size d is the diameter of a circle that has the same area as the particle’s 2D projection. To analyze the PSD, particle sizes were distributed into 82 size classes from 5 µm to 10,000 µm, equally spaced on a logarithmic scale, to obtain a reasonable resolution. Volumetric cumulative distributions (Q_3_) were extracted from PAQXOS and probability density distributions (q_3_) were obtained by derivation of Q_3_. A bimodal Gaussian fit (R^2^ > 0.85 in all treated cases) according to Equation (1) was applied to the q_3_ distributions.
(1)$$q_{3}\left(d\right)\approx \frac{A_{f}}{\sigma_{f\sqrt{2\pi }}}e^{\frac{-{\left(d-\mu_{f}\right)}^{2}}{2\sigma_{f}{}^{2}}}+\frac{A_{c}}{\sigma_{c\sqrt{2\pi }}}e^{\frac{-{\left(d-\mu_{c}\right)}^{2}}{2\sigma_{c}{}^{2}}}$$
where $\sigma_{f}$, $\sigma_{c}$ is the standard deviation; $\mu_{f}$, $\mu_{c}$ is the Gaussian mean particle size; and $A_{f}$, $A_{c}$ is the Gaussian peak probability of the fine and coarse particles, respectively. Peak probabilities were utilized to define the fraction of fines $\omega_{f}$ according to Equation (2).
(2)$$\omega_{f}=\frac{A_{f}}{A_{f}+A_{c}}$$

The mode with lower particle sizes is thus assumed to represent the fines, whereas the second mode, if present, represents coarse agglomerated particles. For the initial distribution, the fines fraction is assumed to be equal to unity. The decrease in fines from initial to the end of the short screw can thus be interpreted as an extension of agglomeration.

Further, two descriptors for particle shape were measured: sphericity $\Psi_{S}$ and aspect-ratio $\Psi_{A}$ [33]. Sphericity is defined as the ratio of diameters of a perimeter-equivalent circle to the real particle. It is a value between 0 and 1, where particles tend to be more irregularly shaped as sphericity approaches zero. The aspect ratio is defined as the ratio of minimum to maximum Feret diameter. This measures particle elongation, where perfectly spherical shapes will have an aspect ratio of 1 and needle-like shapes will approach 0.

### 2.5. Residence Time Distribution

Residence time distribution was measured after reaching steady state conditions by adding 2 mg of the color-dye Tartrazine (less than 1 wt% relative to solid throughput per second) in a short-duration pulse. Camera measurements were utilized according to the method proposed by Kumar et al. [34]. A Python script was developed to postprocess the videos. The video was processed frame-by-frame to obtain mean color saturation of particles in a region of interest, see Figure 2. As RGB color space is device-dependent, it was converted to device-independent CIELab color space with color coordinate L*, a* and b*. To measure the concentration of tracer in the particles, the mean a* value of every frame in the region of interest was utilized. The resulting curve is normalized to obtain an exit age probability E(t) as a function of time t by Equation (3), where $C_{i}$ and Δt refer to the mean a* value of the i-th frame and the time interval between frames, respectively.
(3)$$E\left(t\right)=\frac{C}{\int_{0}^{\infty }C\mathrm{dt}}=\frac{C_{i}}{\sum C_{i}\Delta t}$$

The curve was fitted by the continuously stirred tank reactor model (nCSTR) using the rtdpy package [35]. Two parameters, namely τ and n, were adjusted to approximate E(t) according to Equation (4).
(4)$$E\left(t\right)\approx \frac{1}{\tau }{\left(\frac{t}{\tau }\right)}^{n-1}\frac{n^{n}}{\left(n-1\right)!}e^{\left(\frac{-\mathrm{nt}}{\tau }\right)}$$
Here, τ determines the mean residence time of all CSTRs and n is the number of CSTRs in series. Furthermore, a plug flow reactor (PFR) model was convoluted with the nCSTR model to determine the time of initial tracer detection at the TSG outlet and, additionally, t > 0 can then be measured starting from PFR seconds after dye insertion. Mean residence time (MRT) of the TSG can thus be approximated according to Equation (5).
(5)MRT≈τ+PF

To validate the camera-based residence time measurement, camera measurements were compared to UV/VIS (Cary 60 Spectrophotometer, Agilent, Santa Clara, CA, USA) dye concentration measurements of samples that were regularly retrieved at the outlet. During selected trials of the DoE, samples of granules were collected for a duration of 5 s at the end of the conveyor belt. Sampling points were from 10 to 60 s in intervals of 10 s. Drawn samples were weighed and diluted in 25 mL of demineralized water. A 0.22 µm filter was used to separate undissolved MCC particles from the solution and, thereby, avoid measurement distortions. The filtrate absorption was spectroscopically analyzed at 425 nm in triplicates. UV/VIS and camera measurements result in different quantities, but, owing to sample collection at fixed time points after dye insertion, the shape of RTD was compared for correlation. Plotting the probability density saturation found in the camera measurements against the mass concentration of Tartrazine found in samples by UV/VIS resulted in good correlations (R² = 0.9957 for N10; R² = 0.9749 for N18) of RTD shape, as indicated by Figure 2.

## 3. Results and Discussion

In the following, the main findings of the DoE are described and discussed to investigate the effect of several critical process parameters for future validation of simulation models.

Adding a DFS element had little to no significant effect on the average particle size (d_50_), which is in line with the findings of Verstraeten et al. [15]. Moreover, there was no evidence that the tooth gap size of DFS elements limits the maximum particle size. However, it is of note that a significant influence of DFS on particle size was strongly dependent on the chosen evaluation parameters, e.g., evaluation with partial least squares regression or multiple linear regression, inclusion of quadratic interactions, and alpha = 0.05 versus alpha = 0.01. Furthermore, compared with L/S, the influence of DFS on d_50_ is small. To realize the beneficial repetitive screw design and owing to the questionable influence of DFS on d_50_ with the current formulation and design space, solely results with no DFS are considered. L/S had the greatest influence on d_50_ and L/S, whereas SFL and screw length had the greatest influence on MRT. Thus, we will focus on these parameters in the following sections. Modde model parameters for d_50_ (Q^2^ = 0.778, R^2^ = 0.869) and MRT (Q^2^ = 0.907, R^2^ = 0.94) show satisfactory model significance with the mentioned factors.

### 3.1. Particle Size and Shape Distributions

Center point experiments indicate a good reproducibility of results. The d_50_ coefficient of variation for center level experiments without DFS was 9.3% for the short and 5.6% for the long screw setup with comparable size distributions. For all SFL and L/S combinations, the PSDs measured for the short screw show significant particle growth, owing to agglomeration after initial wetting and kneading; see Figure 3. For high L/S, the PSD range is wide as fines and highly agglomerated particles are present. As expected, particle diameters become larger for high L/S. Contrary to expectation, there is little to no substantial growth of particles visible between the short and long screw.

Both short and long screws exhibit bimodality at low L/S, which reduces to unimodality as L/S and SFL increase. Both curves have a sharp cutoff for oversized particles and a long tail for particles smaller than the average at all parameter combinations. Hence, few oversized particles and many fines are found. At high L/S, the amount of fines is higher for low SFL than for high SFL, because the PSDs are shifted towards low particle sizes. At high L/S, agglomeration seems more dominant than attrition and breakage; therefore, the higher amount of fines at low SFL could be due to less agglomeration than with high SFL. However, for low L/S, it is the other way around. For low L/S, the PSDs are shifted more towards low particle sizes for high SFL than for low SFL. Therefore, at low L/S, agglomeration is more dominant at low SFL, which could be due to better liquid homogenization. Hence, the effects of SFL at high and low L/S are contradictory for its effect on particle size, emphasizing the complex interplay between different granulation processes (breakage, agglomeration, attrition, and so on). It is important to note that the trends for high L/S are more stable than for low L/S. It seems that a higher liquid content lowers the influence of several agglomeration impeding granulation phenomena, emphasizing a process stabilizing effect of higher L/S.

For most of the parameter combinations, the amount of fines decreases with increasing screw length, while the mean coarse particle size remains nearly constant or increases, except for low L/S and low SFL; see Figure 4. Thus, a long screw does not appear to further benefit granulation for the present formulation design space. Large specific shear forces for low SFL as well as local accumulations and attrition inside kneading zones could lead to an increase of fines because of higher friability at low L/S. The center point also shows an increase of fines, but at the same time, coarse particle’s d_50_ increases, which shows a superimposed effect of agglomeration and breakage or attrition. As a peristaltic pump was utilized, the liquid dripped into the barrel. Thus, it is not evident if attrition of especially large particles due to decreasing SFL or the inability to agglomerate particles due to poor liquid distribution—see Vercruysse et al. [23]—is causing a decreasing agglomeration rate for the long screw. However, for PSDs where the long screw curve is shifted to the left, i.e., to lower particle sizes, it is evident that attrition and breakage was the cause. Contrary to the findings of Vercruysse et al., when additionally varying SFL in the design space, fast granulation kinetics are observed. The second kneading section does not significantly contribute to agglomeration of particles. Most often, the mean particle sizes are lowered or constant during second kneading, which could be due to breakage and attrition in the long screw. Nevertheless, it is important to note that, for low L/S and high SFL, both less fines and more oversized particles were produced, emphasizing the importance of both parameters and the complex interplay between agglomeration, breakage, and attrition in the process. Furthermore, screw configurations in Vercruysse et al. differed from the one used in this study and are thus hardly comparable. Moreover, findings of Verstraeten et al. [15] about the necessity to include two kneading compartments for better liquid homogenization to allow further granulation of hydrophilic formulations cannot be confirmed as granulations kinetics seem to be faster than expected. As mentioned, scale and formulation of the investigations make results difficult to compare and it is not evident if other effects are superimposed, which emphasizes the need for an accurate simulation model that can account for particle and liquid interactions with screw geometries.

The shape of particles does not significantly change for different parameter combinations. On average, the particles have a spherical shape with small irregularities; see Figure 5. It is of note that no porosity measurements were performed and, therefore, it is not evident whether porosity changes while shape parameters stay the same. However, sphericity and aspect ratio were chosen given that it is unlikely that both remain unchanged while particles are deformed. Particles below 20 µm do not change shape during granulation and have higher sphericity while being more elongated compared with other particles. Moreover, some very small and spherical particles exist. These particles could either be ungranulated or generated owing to attrition of larger particles. In general, particle sizes with a few particles are more non-spherical and irregularly shaped than the rest. At the first sampling point, the sphericity shows a descending right tail, due to a lack of particles at corresponding particle sizes. Small particles are elongated, but remain at a high sphericity throughout granulation, while large particles have an equal elongation, but a lower sphericity. Aspect ratio as well as sphericity significantly increase between the first and second sampling points for particles larger than 20 µm. Kneading particles in the first section are thus beneficial for a spherical shape. The second kneading section, however, does not clearly change the shape. Between the second and third sampling points, the particles barely changed shape. Hence, the long screw has no obvious influence on shape when compared with the short screw.

The mentioned findings on PSDs, fines, and particle shapes question the necessity of applying a second kneading zone for this formulation. However, porosity measurements would help to support the assumptions of the underlying effects. Analyses on tabletability and tablet properties might reveal practical relevance for continuous manufacturing of tablets.

### 3.2. Residence Time Distributions (RTDs)

All RTDs show a significant initial peak. After reaching peak saturation, the dye concentration in the particles, and thus also the visible color saturation, exponentially decrease. For varying SFL and L/S, the MRT changes; see Figure 6. Low SFL and L/S has the narrowest RTD, while it gets wider for higher SFL and L/S ratio, thus increasing MRT.

For low L/S, the RTD is smooth and does not oscillate, while for high L/S, the signal exhibits a high frequency oscillation that gets dampened as SFL increases. Meier et al. [12] recently found pressure fluctuations in front of kneading zones at low SFL. They hypothesize that these fluctuations occur owing to material blockage in front of kneading sections. Blockages could lead to dye being successively held up from flowing freely through the barrel. Therefore, a pulse with higher dye concentration from particles that were blocked for a short period follows a pulse of lower dye concentration. Although different measurement techniques, screw geometry, as well as formulations were utilized, the observed phenomenon could be identical to what was reported in Meier et al. and important regarding process stability. A study by Zheng et al. [26] supports the findings of Meier et al., as it shows that single particles can be stuck inside the kneading sections while flowing through the barrel, even without liquid. As continuous residence time measurement is already developed as a process analytical technology and easy to handle, it could be superior to pressure sensors for process stability analysis. However, this hypothesis still must be verified by combined RTD and continuous mass flow measurements.

The nCSTR fit parameter n, which describes the skewness of the RTD curve, varies in a narrow range between 1.14 and 2.3 with a low standard deviation of 0.65, and thus is negligibly affected by design factors. RTD fits, therefore, primarily vary regarding τ and PFR, which approximates MRT according to Equation (5). MRT can thus be modeled without taking n into account. A correlation is found when a bilinear fit is applied according to Equation (6).
(6)$$\mathrm{MRT}=N_{\mathrm{Segments}}\;\left(a\;\mathrm{SFL}+b\;L/S+c\right)$$
Here, a, b, and c are polynomial coefficients. The bilinear fit of MRT can be equally applied to both the short and long screw when normalized by the number of repetitive screw segments $N_{\mathrm{Segments}}$ (short = 1, long = 2), implying that doubling the screw length doubles the MRT; see Figure 7. The latter also implies that no further evolution in powder flowability takes place in the second section of the long screw, which is consistent with the above findings on particle size and shape evolution. Therefore, SFL and L/S are suitable parameters to model MRT. Center point experiments show relatively large residuals to the bilinear fit. A quadratic fit made predictions more accurate and aligned with the center point experiments. However, although R^2^ increases, Q^2^ decreases when applying a quadratic fit, which indicates over-fitting. For low SFL and L/S, the number of screw segments has less influence on MRT. Deviations between long and short screws are small. Thus, conveying of powder for low SFL and L/S seems to be dominant over viscous, kneading, and mixing effects. It is of note that a hydrophobic formulation might behave differently as opposed to the current hydrophilic formulation. Measurement errors of continuous camera measurements also have an impact on the accuracy of results.

In RTD of color saturation probability, peaks were found well beyond initial peaks, e.g., in Figure 6 for high L/S and high SFL. Second peaks were solely found for a long screw setup and did not appear for low L/S and low SFL, although they could be very small and indistinguishable from background noise of measurements. The probability for second peaks increased with higher MRT and they seem to be related to material blockage. However, parameter characteristics for SFL and L/S are contradictory, which suggests a second phenomenon. Furthermore, second peaks also occur for non-oscillating camera measurements. While material blockage, as reported by Meier et al., seems to show high frequency oscillations, this second phenomenon has low frequency oscillations. However, a Fourier transformation of RTDs was not able to characterize both high and low frequency oscillations. The peaks could indicate large mass flow pulsations where material is stuck at some location of the barrel, detaches, and leaves the granulator with a dye concentration from an earlier point in time, which leads to a high peak of saturation in camera measurements. Video analysis suggests that peaks in RTDs are related to mass flow pulsations, but do not show where material accumulates. These mass flow pulsations are more visible for experiments with high L/S and in particular in videos of center point experiments; see Figure 8 for a possible pulsation in experiment N21. Assuming this hypothesis is true, mass flow pulsations are related to material hold-up H, which is the average mass of particles and fluid in the barrel under steady-state conditions and assuming no dead zones, defined by Equation (7).
(7)$$H=\dot{m}\;\mathrm{MRT}$$

As MRT is a function of L/S and SFL and screw length is also considered with this parameter, we can characterize the mass flow pulsations solely by hold-up. Moreover, the pulsation strength P_r_ can be quantified as the relative height of the second to the first peak in the RTD. A logistic function according to Equation (8) shows good correlation to quantified data; see Figure 9.
(8)$$P_{r}\approx \frac{L}{1+e^{-k\left(x-x_{0}\right)}}$$

Here, L, k, and $x_{0}$ are fit parameters. Relative peak height was manually quantified and not by an automatic peak detection. Still, the sample root mean squared error, being 0.00624, is sufficiently small. Figure 9 indicates that, for the present set-up, process instabilities are to be expected when hold-up surpasses 14.2 g. Deviations from the logistic function can be explained by interferences in camera measurements, manually determining relative peak height and fluctuations in powder feeder mass flow.

Hold-up increases with higher MRT and so does the pulsation strength. For the present design, short screws as well as low L/S and SFL seem to be beneficial in terms of process stability. However, SFL should not be decreased to a level where accumulation and attrition in kneading sections would take place. Moreover, hold-up could be interpreted as a metric for powder flowability. Highest pulsations occur for large SFL and L/S. Flowability is lowered by both compaction due to higher pressure and high apparent viscosity of the wet bulk. It remains to be seen if other formulations show identical behavior. If not, material hold-up and thus the probability to encounter mass flow pulsations might be correlated with the Hausner-ratio of different formulations.

If these pulsations occur by mass flow fluctuations, a continuous catch scale measurement will outperform continuous RTD measurements. Both measurement oscillations and mass flow pulsations can be underlined by combining continuous RTD and mass flow measurements in future research. It is also not evident whether only very high fluctuations where measured and lower frequency fluctuations are not visible by camera measurements as merely the mean saturation in the region of interest is calculated for every frame. For continuous pharmaceutical manufacturing, these large mass flow pulsations could be problematic as a certain process instability and, thereby, deviating product quality could be expected, e.g., negatively effecting mass and content uniformity of the generated dosage forms. To avoid the described kinds of process disturbances, granulation parameters should be carefully chosen, preferring a smaller hold-up.

## 4. Conclusions

A two-level full factorial design of experiments (DoEs) with a repetitive screw setup was carried out to measure the effect of specific feed load (SFL), liquid-to-solid ratio (L/S), and inclusion of a distributive feed screw (DFS) on particle size and shape as well as residence time distribution (RTD). An intermediate sampling point was obtained by changing inlet ports from a long screw with two kneading sections to a short screw with one kneading section. To measure RTDs, the color saturation of granules was continuously monitored using a UV/VIS validated camera measurement. Particle size analysis was performed by dynamic image analysis (QICPIC).

DFS elements had little to no effect on particle size distributions (PSDs) and were thus omitted from the results. Particle size analysis indicated fast granulation kinetics. Neither in particle size nor in shape was a significant change observed when changing from short to long screw configuration. Moreover, a large increase in fines was found for low L/S and SFL, which indicates increased breakage and attrition with two kneading sections. Mean residence time was found to correlate well with L/S and SFL and a bilinear fit was considered suitable. Furthermore, mass flow pulsations, undesirable for continuous manufacturing, were identified in RTD measurements by observing a second peak. The magnitude of mass flow pulsations was found to correlate with a logistic function of material hold-up. In summary, fast granulation kinetics and process instabilities for long screws indicate that, contrary to current standards, one kneading section might be sufficient for granulation of hydrophilic lactose/MCC-based formulations. However, RTD should be combined with continuous mass flow measurements to validate findings in future research and to evaluate process stability.

Still, DoEs are prone to contradictory findings as they are dependent on screw configurations and formulations. As mentioned earlier, a comprehensive multiscale model to account for particle interactions in twin-screw wet granulation is still missing. The performed DoE is beneficial for simulation purposes and thus should be utilized to further develop the proposed simulation framework to include both formulation as well as configuration dependency and possibly transition from real to virtual design spaces.

## Figures and Tables

**Figure 1 pharmaceutics-13-00645-f001:**
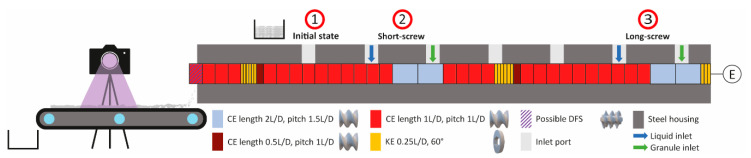
Schematic representation of the experimental setup with (from right to left) engine, twin-screw granulation (TSG) screw, belt conveyor, camera for residence time distribution (RTD) measurements, and sample container. Red circles indicate the data sampling points. DFS, distributive feed screw; CE, conveying element; KE, kneading element.

**Figure 2 pharmaceutics-13-00645-f002:**
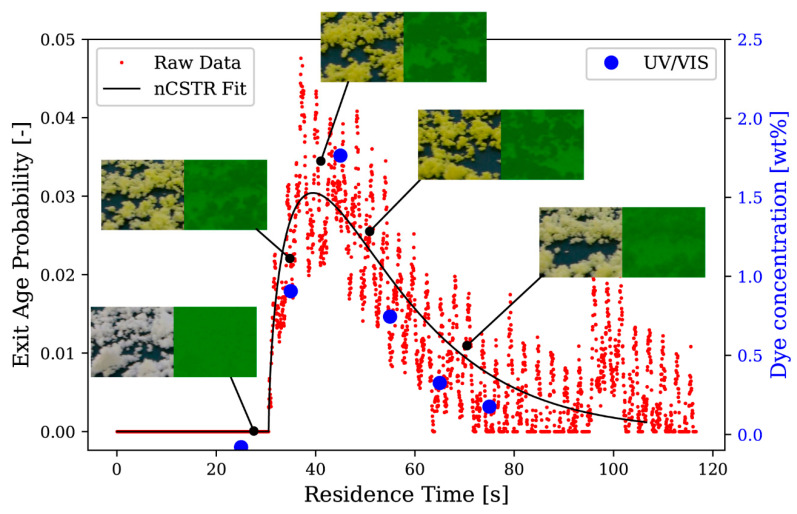
Schematic overview of video frame evaluation in a region of interest and correlation of volumetric probability distribution (q_3_) by continuous camera measurements (red dots) and sample Tartrazine ratio by UV/VIS spectral measurements (blue dots). nCSTR, continuously stirred tank reactor model.

**Figure 3 pharmaceutics-13-00645-f003:**
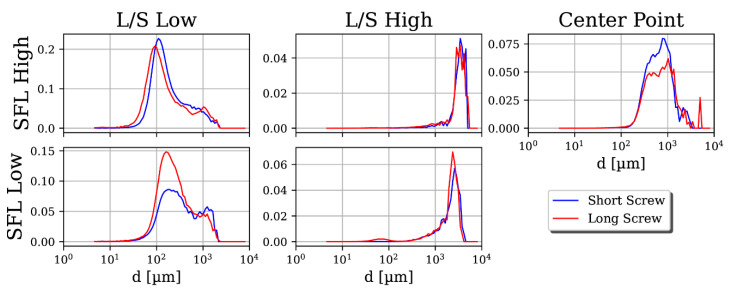
Volumetric probability density size distributions (q_3_) in a long and short screw setup for all combinations of low and high specific feed load (SFL) and liquid-to-solid ratio (L/S) parameters as well as a center point.

**Figure 4 pharmaceutics-13-00645-f004:**
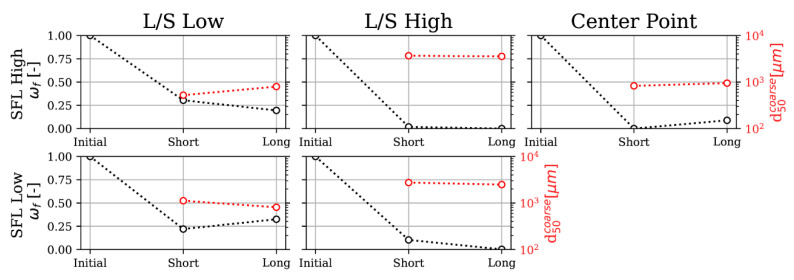
Fraction of fines and d_50_ of coarse particles calculated by a bimodal Gaussian fit over screw length for all combinations of low and high SFL and L/S parameters and a center point.

**Figure 5 pharmaceutics-13-00645-f005:**
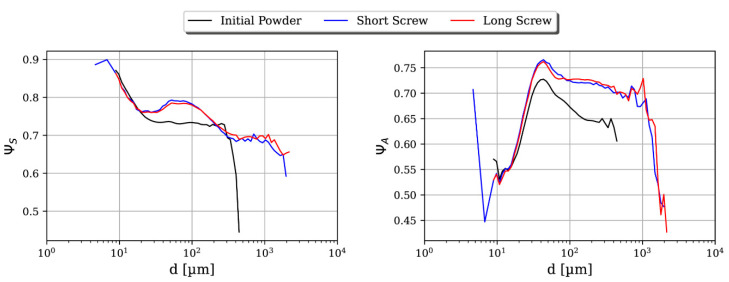
Sphericity and aspect ratio distributions over particle size for low L/S and low SFL for the initial powder, the short screw, and the long screw configuration.

**Figure 6 pharmaceutics-13-00645-f006:**
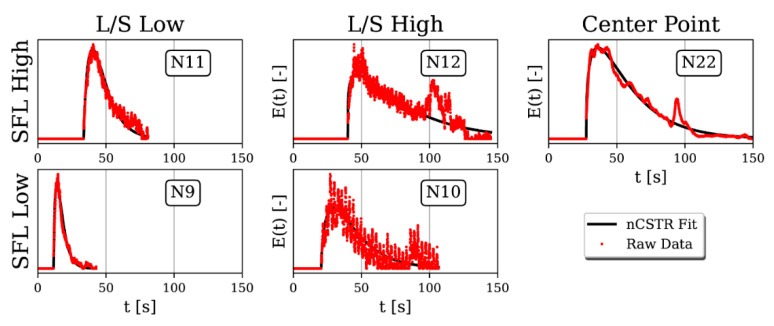
Volumetric probability density distributions (q_3_) of continuous color saturation camera measurements (red dots) approximated by a nCSTR fit (black line) for varying SFL and L/S on a long screw.

**Figure 7 pharmaceutics-13-00645-f007:**
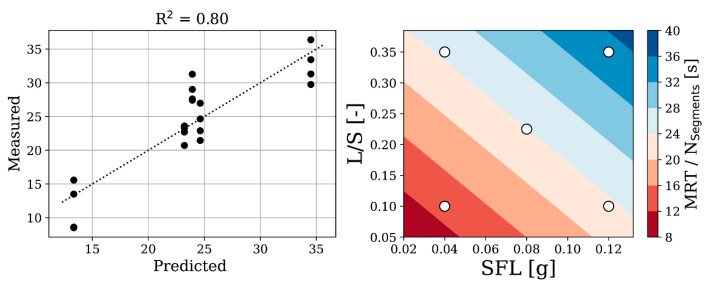
Left: Bilinear fit of $\mathrm{MRT}/N_{\mathrm{Segments}}$ (a = 123.46 s/g, b = 45.16 s, c = 3.89 s) with correlation plot between predicted and measured $\mathrm{MRT}/N_{\mathrm{Segments}}$. Right: surface plot of predicted mean residence time (MRT) per repetitive screw segments. White dots indicate design space parameter combinations that determine predicted MRT for the correlation plot.

**Figure 8 pharmaceutics-13-00645-f008:**
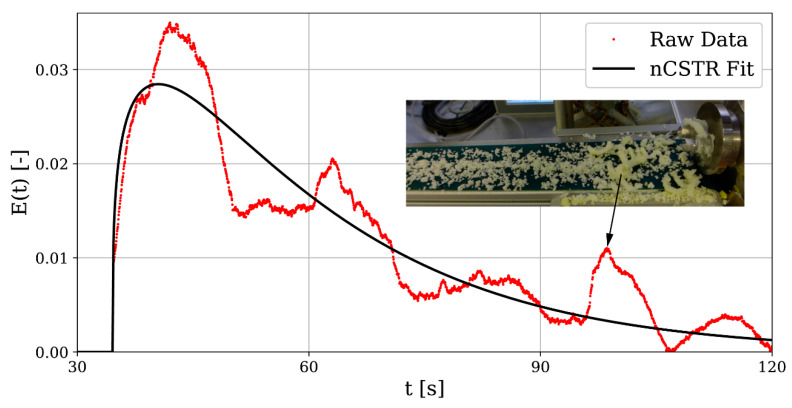
Residence time distribution E(t) for experiment N21, showing eventual mass flow pulsations. Inset shows video snapshot, taken at the time of the second peak.

**Figure 9 pharmaceutics-13-00645-f009:**
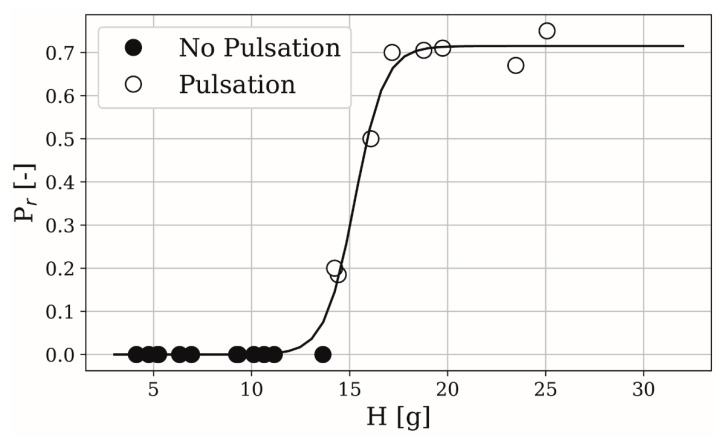
Characterization of mass flow pulsations by fitting a logistic function (L = 0.715, $x_{0}$ = 15.268, k = 1.327), where white markers are experiments with a second peak and black markers without.

**Table 1 pharmaceutics-13-00645-t001:** Composition of the premix.

Component	Grade	Supplier	Fraction [wt%]
Lactose monohydrate	Lactochem^®^ 200M	DFE, Goch, Germany	85
Microcrystalline cellulose	Avicel^®^ PH101	FMC, Brussels, Belgium	10
Polyvinylpyrrolidone	Kollidon^®^ 25	BASF, Ludwigshafen, Germany	5

**Table 2 pharmaceutics-13-00645-t002:** Two-level full factorial design of experiment (DoE) parameter space. L/S, liquid-to-solid ratio; SFL, specific feed load. DFS, distributive feed screw.

Parameters	Low Level	Center Level	High Level
L/S [-]	0.1	0.225	0.35
SFL [g]	0.04	0.08	0.12
Screw length	short	short/long	long
DFS	w	w/w/o	w/o

## Data Availability

The data presented including all tools for data analysis are available in a publicly accessible repository of 4TU.ResearchData: https://www.doi.org/10.4121/14248433.

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
