# Peer review of "Parametric Study of Residence Time Distributions and Granulation Kinetics as a Basis for Process Modeling of Twin-Screw Wet Granulation"

_pharmaceutics, 2021, doi:10.3390/pharmaceutics13050645_

Round 1

Reviewer 1 Report

This is a very interesting application of PAT/QbD approach. Solid work. Congrats! Just one remark - couldn't get to the data source in the public repository. Is it a mistake in the address?

Author Response

Thank you very much for your time to review our manuscript and the nice comments.
Concerning your remark: We did not upload the dataset just yet. We were waiting for a first positive round of reviewing. I will upload the dataset as soon as i have revised the first version, which will approximately be done in max. 5 days. Then it should be accessible.
For now this is the (still empty) repository: https://figshare.com/s/118def92ab1591fdc309

kind regards

Timo Plath

Reviewer 2 Report

GENERAL COMMENTS: The manuscript is generally well written and quite clear to understand.

SPECIFIC COMMETS:

L98. “A promising solution is to develop a comprehensive DPM-PBM-CFD heterogeneous multiscale model” Such a model combines many “moving parts” and needs to be explained in greater detail.

L101. “The focus of this paper is on executing a DoE that is not only designed to optimize a specific screw setup or to get further process insight, but also allows calibration and validation of the proposed simulation framework specifically with regard to residence time and particle size predictions.” Please state the objective in a simpler manner. I found myself reading this sentence repeatedly, trying to grasp its full meaning.

L152. The sentence is incomplete.

Figure 2. Explain how the different data types were combined into a single plot. Specifically, the vertical scaling appears arbitrary.

L280. A hyperlink error is present.

Figure 7. I expect the labels “measured” and “predicted” to be swapped. If not, explain why you would obtain so few levels of predicted values for a range of measured ones.

Figure 9. Use of an R2 value for a non-linear model is misleading, especially one where most values are constant at upper or lower levels.

Author Response

Thanks for the time you took to review! Your comments really helped to improve our manuscript.

I will answer each of your comments point-by-point in the following.

A minor spell check was performed to improve English language and style.

L98. “A promising solution is to develop a comprehensive DPM-PBM-CFD heterogeneous multiscale model” Such a model combines many “moving parts” and needs to be explained in greater detail.

That's a good point! The paragraph was reworked and we hope it is explained in enough detail now. there would of course be more to talk about, but as this is not the main topic we want to keep it as detailed but also as short as possible. Future publications will give more insight on this.

L101. “The focus of this paper is on executing a DoE that is not only designed to optimize a specific screw setup or to get further process insight, but also allows calibration and validation of the proposed simulation framework specifically with regard to residence time and particle size predictions.” Please state the objective in a simpler manner. I found myself reading this sentence repeatedly, trying to grasp its full meaning.

Thanks for pointing this out. The sentence was split into two to make it more understandable. We hope that the full meaning will get clearer in this way and that the whole construction of the sentence is more neatly integrated.

L152. The sentence is incomplete.

Thanks for pointing it out. Unfortunately we did not find an incomplete sentence at L152. However at L150 we found a sentence which was unclear: "Therefore, to better represent the process samples were oven-dried (FDL 115, Binder, Germany) at 40 °C at ambient humidity for further investigations."
We added a comma after "process" to help understanding the construction of the sentence.

Figure 2. Explain how the different data types were combined into a single plot. Specifically, the vertical scaling appears arbitrary.

Thanks for the comment. The vertical scaling is indeed arbitrary and by looking over it a mistake was noticed. One point of the UV/VIS measurement was missing and is now added to the figure. The dye concentration (i.e. the tartrazine ratio in the sample) measured by UV/VIS can not be directly compared to the exit age probability of the camera measurements, because the latter does not result in absolute values as it depends e.g. on the exposure of samples, camera configurations, etc. But we took samples at fixed points in time which makes it able to at least compare the shape of UV/VIS RTD with the camera RTD. A further clarifying sentence was added.

L280. A hyperlink error is present.

Thanks for pointing that out. The issue was solved.

Figure 7. I expect the labels “measured” and “predicted” to be swapped. If not, explain why you would obtain so few levels of predicted values for a range of measured ones.

Thanks for the comment. The Measured values are obviously obtained directly from the experiments and therefore we have a range of measured MRT values. However, for the predicted values we see 5 measurement points (white markers) on the right figure which represent the DoE points. These points have a fixed RTD value which result in the five mentioned levels in the left figure. So basically the levels represent the white markers on the right figure. I hope this explanation is satisfactory and understandable. The caption was updated to clarify it further.

Figure 9. Use of an R2 value for a non-linear model is misleading, especially one where most values are constant at upper or lower levels.

That is an important insight we overlooked. Thanks for pointing that out!
We recreated the figure without R2. Also we checked the sample root mean squared error whose value we added to the text in another clarifying sentence. Even if the constant values at the left tail (black dots except the last one) are not taken into account the RMSE= 0.0277 is still negligibly low.